# Elegant and Innovative Recoding Strategies for Advancing Vaccine Development

**DOI:** 10.3390/vaccines13010078

**Published:** 2025-01-16

**Authors:** François Meurens, Fanny Renois, Uladzimir Karniychuk

**Affiliations:** 1Centre de Recherche en Infectiologie Porcine et Avicole, Faculté de Médecine Vétérinaire, Université de Montréal, Saint-Hyacinthe, QC J2S 2M2, Canada; fanny.renois@umontreal.ca; 2Department of Veterinary Microbiology and Immunology, Western College of Veterinary Medicine, University of Saskatchewan, Saskatoon, SK S7N 5E2, Canada; karniychuk.1@osu.edu; 3Department of Veterinary Biosciences, College of Veterinary Medicine, The Ohio State University, Columbus, OH 43210, USA

**Keywords:** genome, recoding, live-attenuated vaccines

## Abstract

Recoding strategies have emerged as a promising approach for developing safer and more effective vaccines by altering the genetic structure of microorganisms, such as viruses, without changing their proteins. This method enhances vaccine safety and efficacy while minimizing the risk of reversion to virulence. Recoding enhances the frequency of CpG dinucleotides, which in turn activates immune responses and ensures a strong attenuation of the pathogens. Recent advancements highlight synonymous recoding’s potential, offering improved genetic stability and immunogenicity compared to traditional methods. Live vaccines attenuated using classical methods pose a risk of reversion to virulence and can be time-consuming to produce. Synonymous recoding, involving numerous codon alterations, boosts safety and vaccine stability. One challenge is balancing attenuation with yield; however, innovations like Zinc-finger antiviral protein (ZAP) knockout cell lines can enhance vaccine production. Beyond viral vaccines, recoding can apply to bacterial vaccines, as exemplified by modified *Escherichia coli* and *Streptococcus pneumoniae* strains, which show reduced virulence. Despite promising results, challenges like ensuring genetic stability, high yield, and regulatory approval remain. Briefly, ongoing research aims to harness these innovations for comprehensive improvements in vaccine design and deployment. In this commentary, we sought to further engage the community’s interest in this elegant approach by briefly highlighting its main advantages, disadvantages, and future prospects.

Recently, an intriguing study by Schön and collaborators has renewed interest within the scientific community in using recoding as a strategy for developing new, safe, and more immunogenic vaccines against infectious diseases [1]. In this comment paper, we aim to outline the key advantages, disadvantages, and prospects of recoding strategies for vaccine development. This commentary has been crafted as a concise narrative review [2], utilizing relevant and recent publications identified through tools like PubMed, Scopus, and Web of Science. We acknowledge that our reference selection may be subject to discussion, but the concise format of this commentary limits our ability to include all references in the field. In their study, Schön and collaborators detail the development and testing of a novel, safe, effective, and adaptable live-attenuated severe acute respiratory syndrome coronavirus 2 (SARS-CoV-2) vaccine, utilizing one-to-stop (OTS) genome modifications to mitigate disease and transmission. The researchers engineered two live-attenuated vaccine (LAV) candidates, OTS-206 and OTS-228, employing the elegant OTS method [3], which involves introducing synonymous codon changes to increase the likelihood of premature termination codons, thereby reducing viral fitness. They also introduced additional modifications such as disabling Nsp1 translational repression [4], removing open reading frames (ORFs) 6, 7ab, and 8 [5,6,7,8], and optimizing the spike polybasic cleavage site [9] to further enhance safety and immunogenicity. Preclinical studies in animal models (mouse and golden Syrian hamster) revealed that OTS-228 demonstrated an optimal safety profile, exhibiting no detectable side effects or transmission in preclinical animal models. Moreover, a single-dose vaccination induced sterilizing immunity in vivo against the homologous wild-type SARS-CoV-2 challenge infection and provided broad protection against Omicron variants BA.2, BA.5, and XBB.1.5, with significantly reduced transmission. Critically, OTS-228 completely prevented transmission to contact animals. In contrast, OTS-206, while highly protective, exhibited some degree of transmission. The superior safety and efficacy of OTS-228, coupled with its lack of transmissibility, makes it a particularly promising candidate for the new generation of vaccines against SARS-CoV-2. This research underscores again the potential of OTS-modified LAVs as a robust and very promising strategy for vaccine development against emerging and re-emerging viral threats. The successful attenuation and enhanced protective efficacy of OTS-228, particularly its inability to transmit, suggest its promise for clinical trials and eventual deployment.

Traditional methods for creating live-attenuated vaccines—one of the oldest types of vaccines [10]—typically involve using naturally occurring strains of microorganisms that have been weakened through serial passages in cell cultures, alternative hosts, or targeted amino acid mutations and deletions. While live-attenuated legacy vaccines, for example, the yellow fever virus 17D vaccine, showed and keeps showing excellent performance, traditional methods can lead to serious safety concerns, including the risk of reversion to virulence, as observed with the poliovirus vaccine [11]. Moreover, the traditional method with blind passages does not allow rapid vaccine generation during outbreaks and epidemics.

As illustrated with this recent study [1], the development of vaccines through recoding strategies has emerged as a promising approach to enhance safety and efficacy while minimizing the risk of reversion to virulence [12,13]. For safety, historical live-attenuated vaccines often rely on several or even one amino acid change, which may compromise safety. In contrast, the synonymous recoding approach implies hundreds of altered codons or dinucleotides, or combinatory approaches, which increase the stability and safety of vaccines. Recoding, particularly synonymous recoding, involves altering the nucleotide sequences of microbial genomes without changing the encoded amino acids. This is the essential advantage for immunogenicity and protective efficacy.

One of the key advantages of synonymous recoding is its potential to increase the frequency of CpG dinucleotides within viral genomes. CpG dinucleotides are naturally underrepresented in vertebrates and in most vertebrate RNA viruses [14,15], although some CpG hotspots can be found in certain regions of various viral genomes [16]. It is believed that single-stranded RNA viruses mimic the CpG composition of their hosts as a strategy to evade detection by the innate immune response [16,17,18]. Host Zinc-finger antiviral protein (ZAP) binds CpG-enriched viral RNA and targets it to exosome degradation, providing attenuation [18,19]. Moreover, the increased number of CpG dinucleotides has been shown to activate innate immune responses via pattern recognition receptors, thereby potentially enhancing the immunogenicity of the CpG-enriched vaccine candidates [20,21]. For instance, single-stranded short RNAs with enriched CpG content activate human monocytes [22]. Moreover, short single-stranded CpG-enriched RNA fragments from the influenza RNA evoke more robust type I IFN and IFN-γ responses in cells [23]. These studies, however, used CpG-enriched short RNA fragments. How CpG enrichment in the viral RNA in the context of an infectious virus affects cellular immune responses remains to be studied. Thompson et al. also highlighted that the enrichment of CpG dinucleotides within the genome of another RNA virus, the influenza A virus, could serve as a strategic approach for LAV development, although it may also impair viral replication and reduce vaccine yield [24]. This trade-off between attenuation and yield is a critical consideration in the design of effective vaccines. The yield, however, can be increased by generating ZAP knockout cell lines, which significantly enhance the titers of CpG-enriched vaccines, making them comparable to those of wild-type viruses [25]. 

Many RNA viruses have also evolved low uracil-phosphate-adenine (UpA) dinucleotide content [26,27]. Enteroviruses enriched for UpA dinucleotides have increased sensitivity to ZAP and attenuated infectious phenotypes in vitro, though to a lesser extent than CpG-enriched variants [26,27,28]. While CpG enrichment without altering UpA content provides attenuation for flaviviruses in vitro and using mouse models [20,25,29,30], simultaneous enrichment of CpG and UpA may jointly result in more robustly attenuated phenotypes.

Moreover, codon deoptimization has been shown to confer genetic stability to attenuated viruses. It can reduce the efficiency of viral protein synthesis and lead to a robust attenuation effect while minimizing the likelihood of reversion to virulence [27,31]. Several examples are available with both RNA and DNA viruses [32,33,34,35,36,37]. The ability to generate genetically stable vaccine candidates is particularly important in the context of rapidly evolving viruses, such as RNA viruses, which are frequently responsible for pandemics [38]. In addition to enhancing safety, recoding strategies can also be combined with other established vaccine approaches to improve efficacy. For example, Udenze et al. suggested that combining CpG recoding with the targeted insertion of sequences for placenta-specific microRNAs could mitigate the risk of transplacental infections, thereby enhancing the safety profile of recoded vaccines [30]. This integrative approach could lead to the development of vaccines that are not only safe but also tailored to specific populations, such as pregnant individuals. 

Besides the development of new vaccines against viruses, the generation of a synthetic *Escherichia coli* genome [39] has enabled new strategies for the elaboration of a new generation of bacterial LAV, as has been illustrated with *Streptococcus pneumoniae* [40]. The recoding of the *S. pneumoniae* pneumolysin gene resulted in the generation of bacterial strains that were less virulent than their wild-type counterparts in mice. Similarly, in *Enterobacteriaceae*, such as *Salmonella enterica* [41], synonymous mutations showed a strong ability to reduce the fitness of the bacteria, and recoding could be advantageously used for vaccine development [13].

Despite the promising results, challenges remain in the practical application of recoding strategies for vaccine development. The phenotypic and genetic stability of recoded vaccine candidates must be thoroughly evaluated to ensure their safety and efficacy in diverse populations [42]. Indeed, while recoding can enhance stability, some studies have reported unexpected phenotypic changes and genetic instability in recoded vaccine candidates. Additionally, the risk of recombination or reassortment between vaccine candidates and wild-type viruses should be considered. Then, the potential for pleiotropic effects resulting from synonymous mutations necessitates the careful characterization of vaccine candidates in preclinical and clinical settings [43]. Regarding immune responses, the introduction of multiple synonymous mutations may also lead to unintended consequences, including altered immunogenicity. More research is needed to fully assess the impact of these changes on immune responses. Additionally, there are challenges in producing recoded vaccines with high yields and consistent quality, as well as in obtaining approval for commercial use.

The future of vaccine development through recoding strategies appears promising [44], with several of the following avenues for exploration: (1) Combination Approaches: integrating recoding techniques with other vaccine modalities, such as mRNA vaccines or viral vector platforms, could enhance safety and efficacy. (2) Tailored Vaccines: Advances in genomics and bioinformatics may enable the design of “personalized” vaccines that are tailored to the specific immune profiles of different populations (see Zika example above). This could be particularly beneficial for vulnerable groups, such as the elderly or immunocompromised individuals. (3) Broader Applications: While much of the current research has focused on viral pathogens, recoding strategies could also be applied, as mentioned above, to bacterial vaccines and other infectious agents. This could expand the utility of recoding techniques in global health initiatives. (4) Most of the studies have been carried out with human pathogens and less with animal pathogens. However, there is an urgent need to develop better vaccines of new vaccines for animal viruses, such as, for instance, the porcine reproductive and respiratory syndrome virus (PRRSV) [45] and the African swine fever virus (ASFV) [46]. The case of PRRSV is quite intriguing. It is a positive single-stranded RNA virus known for its rapid evolution and extremely high mutation rate—it has one of the highest if not the highest, mutation rates among RNA viruses [47]. Vaccination against PRRSV, which relies on inactivated vaccines and live-attenuated vaccines (LAV), has not provided comprehensive and reliable protection against all aspects of the disease and all the strains. Quite recently, in Denmark, the use of live-attenuated vaccines (LAV) was suspended due to recombination events between vaccine strains, which led to a severe outbreak [48,49]. This incident highlights the urgent need for new, safer, and more effective vaccines to combat this significant disease in the pig industry. The urgent need to develop better vaccines for animal viruses should also encompass the development of One Health vaccines aimed at vaccinating livestock to prevent agricultural losses and reduce the risk of zoonotic transmission to humans. (5) Longitudinal Studies: Ongoing research should focus on long-term studies to assess the durability of immune responses elicited by recoded vaccines. Understanding the longevity of protection will be critical for informing vaccination strategies and schedules.

To conclude, recoding strategies represent a significant advancement in vaccine development, offering a pathway to create safer and more effective live-attenuated vaccines. By leveraging techniques such as codon deoptimization, “one-to-stop” codons introduction, and CpG/UpA enrichment, researchers can design vaccines that elicit robust immune responses while minimizing the risks associated with traditional vaccine approaches. Ongoing research and clinical trials will be essential to fully realize the potential of these innovative strategies in combating viral diseases.

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
