# Peer review of "Elegant and Innovative Recoding Strategies for Advancing Vaccine Development"

_vaccines, 2025, doi:10.3390/vaccines13010078_

Round 1

Reviewer 1 Report

Comments and Suggestions for Authors

Accept with minor updation

Recoding strategies represent a breakthrough in vaccine development and offer the potential for safer and more effective live attenuated vaccines (LAVs) by using genetic modifications to improve stability, safety, and immunogenicity. This approach uses synonymous recoding to make large codon changes that increase the frequency of CpG dinucleotides to stimulate innate immune responses while minimizing the risk of reversion to virulence. Innovative methods such as the one-to-stop genome modification (OTS) approach have shown promising results, especially in preclinical studies on SARS-CoV-2, with increased safety and efficacy against different variants. However, the risks associated with developing LAVs for highly variable viruses such as SARS-CoV-2 remain a concern, as mutations make maintaining safety and efficacy significantly more difficult. While OTS technology offers a safer approach, overcoming this complexity requires a clear understanding of the challenges and factors influencing LAV development. In addition, regulatory hurdles specific to recoding strategies must be considered, such as ensuring long-term genetic stability, mitigating unintended mutations, and ethical considerations when tailoring vaccines to particular populations. These challenges underscore the need for comprehensive research and long-term evaluation. Despite these limitations, the integration of recoding with other vaccine technologies and the expansion of its application to human and animal pathogens offers a promising avenue for future vaccine innovation.

Author Response

Recoding strategies represent a breakthrough in vaccine development and offer the potential for safer and more effective live attenuated vaccines (LAVs) by using genetic modifications to improve stability, safety, and immunogenicity. This approach uses synonymous recoding to make large codon changes that increase the frequency of CpG dinucleotides to stimulate innate immune responses while minimizing the risk of reversion to virulence. Innovative methods such as the one-to-stop genome modification (OTS) approach have shown promising results, especially in preclinical studies on SARS-CoV-2, with increased safety and efficacy against different variants. However, the risks associated with developing LAVs for highly variable viruses such as SARS-CoV-2 remain a concern, as mutations make maintaining safety and efficacy significantly more difficult. While OTS technology offers a safer approach, overcoming this complexity requires a clear understanding of the challenges and factors influencing LAV development. In addition, regulatory hurdles specific to recoding strategies must be considered, such as ensuring long-term genetic stability, mitigating unintended mutations, and ethical considerations when tailoring vaccines to particular populations. These challenges underscore the need for comprehensive research and long-term evaluation. Despite these limitations, the integration of recoding with other vaccine technologies and the expansion of its application to human and animal pathogens offers a promising avenue for future vaccine innovation.

Thank you for your positive comments. We introduced various changes in the manuscript (in yellow) to improve it.

The article is now longer (30 lines more – 2018 words total) and includes 16 more references.

Reviewer 2 Report

Comments and Suggestions for Authors

The current manuscript provided a review of the genome recoding strategy for vaccine development. It highlights the advancements of the recoding strategies particularly focusing on synonymous CpG enrichment for live attenuated vaccine (LAV) development against human viral pathogens. While the potential for non-human viral pathogens and bacterial vaccine development using the recoding methods is also discussed by these authors and considered as promising additive to the current strategies. At the end, several topics of further explorations are proposed regarding the safety, efficacy and applications of the recoding LAV platforms.

The manuscript is well written and easy to read, with a clear structure. It will successfully elicit beginner’s or new learner’s interest on the recoding LAV strategies and seek for more on this direction. While it will be bit boring for other people who are already working on the same or similar directions, for the facts/results presented in the manuscript can also be drawn on from other publications (including reviews). I found there is little innovative thinkings or original propositions in the manuscript, which probably due to the nature of the comment paper (topic is focused, size is limiting...). Even for some known critical points such as genetics stabilities, safety issues (recombination or reassortments), and the versatility during the design and applications of the recoding strategies, a deeper discussion/exploration is also lacking. Besides, the authors only mentioned the CpG recoding which is bit misleading since at least both CpG and UpA (in terms of RNA viruses) are critical for attenuating the virulence, and the combinations of both are also promising strategy in making LAV vaccines.

I appreciated the enthusiasm of these authors in projecting a promising platform for vaccine development, while it would be more attractive if these authors could focus on one detailed area/issue/possibility during the recoding LAV development and make targeted propositions, rather than simply presenting the “promising”.

Author Response

The current manuscript provided a review of the genome recoding strategy for vaccine development. It highlights the advancements of the recoding strategies particularly focusing on synonymous CpG enrichment for live attenuated vaccine (LAV) development against human viral pathogens. While the potential for non-human viral pathogens and bacterial vaccine development using the recoding methods is also discussed by these authors and considered as promising additive to the current strategies. At the end, several topics of further explorations are proposed regarding the safety, efficacy and applications of the recoding LAV platforms.

The manuscript is well written and easy to read, with a clear structure. It will successfully elicit beginner’s or new learner’s interest on the recoding LAV strategies and seek for more on this direction. While it will be bit boring for other people who are already working on the same or similar directions, for the facts/results presented in the manuscript can also be drawn on from other publications (including reviews). I found there is little innovative thinkings or original propositions in the manuscript, which probably due to the nature of the comment paper (topic is focused, size is limiting...). Even for some known critical points such as genetics stabilities, safety issues (recombination or reassortments), and the versatility during the design and applications of the recoding strategies, a deeper discussion/exploration is also lacking. Besides, the authors only mentioned the CpG recoding which is bit misleading since at least both CpG and UpA (in terms of RNA viruses) are critical for attenuating the virulence, and the combinations of both are also promising strategy in making LAV vaccines.

Thank you very much for all these helpful comments. We made several changes (in yellow in the text). Also, we clarified the point regarding CpG and UpA (see lines 105-110 and line 178). We also go deeper in the discussion, even if we are clearly limited by the format of a comment paper, see below the recommendations from the journal regarding that type of publication. See for instance, lines 84-88 and 160-169.

“Commentaries should be focused on a new technique or an important/groundbreaking development in the field, that is of importance to the journal and readers. The purpose of these pieces could be to start a wider discussion in the field, a call for action, or dissemination of important information. They are usually no more than 2500 words, usually contain no figures and may focus on current advances as well as future directions of a certain topic”.

The article is now longer (30 lines more) and includes 16 more references.

I appreciated the enthusiasm of these authors in projecting a promising platform for vaccine development, while it would be more attractive if these authors could focus on one detailed area/issue/possibility during the recoding LAV development and make targeted propositions, rather than simply presenting the “promising”.

Thank you for the comment, see our response above.

Reviewer 3 Report

Comments and Suggestions for Authors

The manuscript "Recoding strategies for vaccine development, elegant and 2 promising" has neither qualty, nor quantity, lack of presentation of results clearly, no figures, no table, not even a heading and sub-heading. the complete manuscript is just text with 150 lines. I am not even able to comment on the manuscript. but 100% sure to reject it. this manuscript need comprehesive revision and author need to vide a clear structure to the manuscript.

Author Response

The manuscript "Recoding strategies for vaccine development, elegant and 2 promising" has neither quality, nor quantity, lack of presentation of results clearly, no figures, no table, not even a heading and sub-heading. the complete manuscript is just text with 150 lines. I am not even able to comment on the manuscript. but 100% sure to reject it. this manuscript needs comprehensive revision and author need to vide a clear structure to the manuscript.

Thank for the assessment of our comment paper. Just to clarify the expectation regarding comment paper. “Commentaries should be focused on a new technique or an important/groundbreaking development in the field, that is of importance to the journal and readers. The purpose of these pieces could be to start a wider discussion in the field, a call for action, or dissemination of important information. They are usually no more than 2500 words, usually contain no figures and may focus on current advances as well as future directions of a certain topic”.

The article is now longer (30 lines more) and includes 16 more references.

Reviewer 4 Report

Comments and Suggestions for Authors

In the present commentary, the authors first described a recently published research article (Schon et al., 2024) that elucidated the benefit of genetic recoding for the development of a safe, and effective vaccine against SARS-CoV-2. They then discussed the advantages of genetic recoding of microorganisms as an innovative strategy for enhancing vaccine safety and efficacy. Genome recoding involves the controlled alteration of genetic sequences to generate organisms with modified phenotypes without altering the encoded proteins. This approach not only advances our understanding of microbial interactions with host innate immune responses but also improves vaccine safety by minimizing the risk of reversion to virulence. Additionally, genetic recoding increases vaccine efficiency by enhancing immunogenicity and hence attenuating pathogenicity. Recoding strategies can introduce CpG-rich sequences, which are efficiently recognized by host innate immune sensors such as Toll-like receptor 9 (TLR9) and cGAS-STING sensors, leading to a more robust host immune response. This method has shown potential for both viral and bacterial vaccines, providing a versatile platform for vaccine development.

Advantages of Genetic Recoding

1.     Reduces the risk of reversion to a virulent strain, a problem associated with the traditional approach of generating live-attenuated vaccines. By this approach, we can replace mutation-prone codons with more stable codons.

2.    Eliminates the time-consuming process of virus attenuation required in conventional approaches e.g. long passage in cell culture.

3.    Enhances the efficiency of the vaccine by introducing the sequence that elicits a stronger immune response e.g. by introducing a more CpG-rich sequence.

4.    Reduce the number of dosages as seen in the case of OTS-28 against SARS-CoV-2.

5. Protect against several strains of viruses

6.    Reduce transmission, enhancing its value not just for individual protection but also as a powerful public health tool.

7.    Low side effects

Disadvantages of Genetic Recoding

1.     Recoded organisms may have reduced and weak genetic stability, impacting long-term viability.

2.    Changes in the genetic code may unpredictably influence host immune response outcomes.

3.    Manufacturing recoded vaccines with high yields and consistent quality is a problem.

4.    Degraded easily (by ZAP) so high yield is an issue.

Future Prospects

1.     Genetic recoding can be applied to both viral and bacterial vaccine platforms, expanding its utility.

2.    Recoding approaches can be combined with adjuvants or other vaccine technologies to enhance efficacy.

3.    The technique shows promise for developing vaccines for livestock and wildlife, addressing zoonotic disease threats.

4.    Future research should focus on assessing recoded vaccines' long-term benefits, efficacy, and safety.

In summary, genetic recoding offers a transformative approach to vaccine development, with the potential to overcome safety and efficacy issues associated with traditional methods. While there are obstacles to overcome, this technology represents a significant step toward next-generation vaccines.

Here are a few minor suggestions for the authors-

1.     The title needs improvement. Please change it to better.  

2.    Line 69, “contract” should be “contrast”.

3. Lines 55-56 should be- “The successful attenuation and enhanced protective efficacy of OTS-228, particularly its inability to transmit, suggest its promise for clinical trials and eventual deployment”.

Author Response

In the present commentary, the authors first described a recently published research article (Schon et al., 2024) that elucidated the benefit of genetic recoding for the development of a safe, and effective vaccine against SARS-CoV-2. They then discussed the advantages of genetic recoding of microorganisms as an innovative strategy for enhancing vaccine safety and efficacy. Genome recoding involves the controlled alteration of genetic sequences to generate organisms with modified phenotypes without altering the encoded proteins. This approach not only advances our understanding of microbial interactions with host innate immune responses but also improves vaccine safety by minimizing the risk of reversion to virulence. Additionally, genetic recoding increases vaccine efficiency by enhancing immunogenicity and hence attenuating pathogenicity. Recoding strategies can introduce CpG-rich sequences, which are efficiently recognized by host innate immune sensors such as Toll-like receptor 9 (TLR9) and cGAS-STING sensors, leading to a more robust host immune response. This method has shown potential for both viral and bacterial vaccines, providing a versatile platform for vaccine development.

Advantages of Genetic Recoding

  1. Reduces the risk of reversion to a virulent strain, a problem associated with the traditional approach of generating live-attenuated vaccines. By this approach, we can replace mutation-prone codons with more stable codons.
  2. Eliminates the time-consuming process of virus attenuation required in conventional approaches e.g. long passage in cell culture.
  3. Enhances the efficiency of the vaccine by introducing the sequence that elicits a stronger immune response e.g. by introducing a more CpG-rich sequence.
  4. Reduce the number of dosages as seen in the case of OTS-28 against SARS-CoV-2.
  5. Protect against several strains of viruses
  6. Reduce transmission, enhancing its value not just for individual protection but also as a powerful public health tool.
  7. Low side effects

Disadvantages of Genetic Recoding

  1. Recoded organisms may have reduced and weak genetic stability, impacting long-term viability.
  2. Changes in the genetic code may unpredictably influence host immune response outcomes.
  3. Manufacturing recoded vaccines with high yields and consistent quality is a problem.
  4. Degraded easily (by ZAP) so high yield is an issue.

Future Prospects

  1. Genetic recoding can be applied to both viral and bacterial vaccine platforms, expanding its utility.
  2. Recoding approaches can be combined with adjuvants or other vaccine technologies to enhance efficacy.
  3. The technique shows promise for developing vaccines for livestock and wildlife, addressing zoonotic disease threats.
  4. Future research should focus on assessing recoded vaccines' long-term benefits, efficacy, and safety.

In summary, genetic recoding offers a transformative approach to vaccine development, with the potential to overcome safety and efficacy issues associated with traditional methods. While there are obstacles to overcome, this technology represents a significant step toward next-generation vaccines.

Thank you for the good assessment of our comment paper.

Here are a few minor suggestions for the authors-

  1. The title needs improvement. Please change it to better.  
  2. Line 69, “contract” should be “contrast”.
  3. Lines 55-56 should be- “The successful attenuation and enhanced protective efficacy of OTS-228, particularly its inability to transmit, suggest its promise for clinical trials and eventual deployment”.

Thank you for the careful revision. The title has been changed in “Elegant and innovative recoding strategies for advancing vaccine development”. Then, the two mistakes have been corrected, see line 78 and 62-64.

Reviewer 5 Report

Comments and Suggestions for Authors

Recent studies show that unmethylated CpG dinucleotides in DNA and RNA vaccines are immunostimulatory and have significant endogenous adjuvant activity. In their brief report, the authors outlined strategies for using recoded vaccines with an increased frequency of CpG dinucleotides in pathogen genomes, which activates immune responses and provides strong mitigation of pathogen exposure when using live attenuated vaccines. Appropriately reengineered vaccines may be an effective way to prevent COVID-19, which is associated with new strains of SARS-CoV-2. At the same time, the authors point out the possibility of adverse effects of these generally promising vaccines. The text of the article is concise and quite understandable. I do not see the need to make significant changes to this article. Meanwhile, I have three unprincipled comments:

(1) Abstract. It is advisable for authors to briefly state the purpose of their short scientific narrative.

(2) Please indicate which databases were used and the principles for including and excluding scientific data sources.

(3) Authors may also wish to consider the following link: Digard P, Lee HM, Sharp C, Grey F, Gaunt E. Intra-genome variability in the dinucleotide composition of SARS-CoV-2. Virus Evol. 2020; 6(2):veaa057. doi: 10.1093/ve/veaa057. PMID: 33029383.

Author Response

Recent studies show that unmethylated CpG dinucleotides in DNA and RNA vaccines are immunostimulatory and have significant endogenous adjuvant activity. In their brief report, the authors outlined strategies for using recoded vaccines with an increased frequency of CpG dinucleotides in pathogen genomes, which activates immune responses and provides strong mitigation of pathogen exposure when using live attenuated vaccines. Appropriately reengineered vaccines may be an effective way to prevent COVID-19, which is associated with new strains of SARS-CoV-2. At the same time, the authors point out the possibility of adverse effects of these generally promising vaccines. The text of the article is concise and quite understandable. I do not see the need to make significant changes to this article. Meanwhile, I have three unprincipled comments:

(1) Abstract. It is advisable for authors to briefly state the purpose of their short scientific narrative.

(2) Please indicate which databases were used and the principles for including and excluding scientific data sources.

(3) Authors may also wish to consider the following link: Digard P, Lee HM, Sharp C, Grey F, Gaunt E. Intra-genome variability in the dinucleotide composition of SARS-CoV-2. Virus Evol. 2020; 6(2):veaa057. doi: 10.1093/ve/veaa057. PMID: 33029383.

Thank you very much for all these really helpful comments. We modified the abstract, see lines 28-30 and the beginning of the text, lines 35-41, for the first comment. Regarding point two, we explained our approach, see lines 35-41.

Thank you for the reference. We added it and modified the text in some places (see lines 84-88).

Round 2

Reviewer 2 Report

Comments and Suggestions for Authors

These authors prepared a revised manuscript in which more text was added. These additions explain the scope of the commentary paper better and well adjust readers’ expectations. Compared to the previous version, these authors also discuss more in specific points and make targeted propositions. I have no further comments.